# Geographical Origin Differentiation of Rice by LC–MS-Based Non-Targeted Metabolomics

**DOI:** 10.3390/foods11213318

**Published:** 2022-10-23

**Authors:** Zhanming Li, Mengmeng Tan, Huxue Deng, Xu Yang, Yue Yu, Dongren Zhou, Hao Dong

**Affiliations:** 1School of Grain Science and Technology, Jiangsu University of Science and Technology, Zhenjiang 212100, China; 2Key Laboratory of Fish Health and Nutrition of Zhejiang Province, Zhejiang Institute of Freshwater Fisheries, Huzhou 313001, China; 3College of Light Industry and Food Sciences, Zhongkai University of Agriculture and Engineering, Guangzhou 510225, China

**Keywords:** metabolomics, adulteration, partial least squares discriminant analysis, geographical origin, chemometrics

## Abstract

Many factors, such as soil, climate, and water source in the planting area, can affect rice taste and quality. Adulterated rice is common in the market, which seriously damages the production and sales of high-quality rice. Traceability analysis of rice has become one of the important research fields of food safety management. In this study, LC–MS-based non-targeted metabolomics technology was used to trace four rice samples from Heilongjiang and Jiangsu Provinces, namely, Daohuaxiang (DH), Huaidao No. 5 (HD), Songjing (SJ), and Changlixiang (CL). Results showed that the discrimination accuracy of the partial least squares discriminant analysis (PLS-DA) model was as high as 100% with satisfactory prediction ability. A total of 328 differential metabolites were screened, indicating significant differences in rice metabolites from different origins. Pathway enrichment analysis was carried out on the four rice samples based on the KEGG database to determine the three metabolic pathways with the highest enrichment degree. The main biochemical metabolic pathways and signal transduction pathways involved in differential metabolites in rice were obtained. This study provides theoretical support for the geographical origins of rice and elucidates the change mechanism of rice metabolic pathways, which can shed light on improving rice quality control.

## 1. Introduction

Rice quality can be affected by many factors, including soil, climate, and water source. Rice adulteration in the market is common, which greatly affects the production and sales of high-quality grain [1,2]. In recent years, rice quality analysis and control have become important research fields of food safety supervision. The analysis of rice’s geographical origin is crucial to ensure the high quality of rice and to promote the development of the rice industry [3,4].

Previous studies have shown that many techniques, including spectral technology, stable isotope technology, volatile matter analysis, mineral element analysis technology, and DNA technology, can be used to detect the component differences in rice, providing alternatives for the traceability of agricultural products [5]. Recently, metabolomics technology has been widely applied in the field of agricultural food analysis [6]. Through LC–MS technology, combined with multivariate statistical methods such as principal component analysis (PCA), partial least squares discriminant analysis (PLS-DA), and cluster analysis, the metabolic pathways of differential metabolites in whole grain brown rice before and after milling were analyzed. Eight secondary metabolites are considered potential biomarkers for distinguishing organic rice and conventional rice based on LC–MS non-targeted metabolomics technology [7]. Non-targeted metabolomics based on headspace solid-phase microextraction, gas chromatography, and MS can also be used to identify japonica rice samples from different origins, and the OPLS-DA model presents excellent geographical discrimination ability [8].

At present, rice analysis based on metabolomics focuses on the screening of differential metabolites and the analysis of material changes in processed rice [9,10,11]. Research on the traceability of rice varieties by metabolomics technology needs to be further developed. In this paper, non-targeted metabolomics technology based on LC–MS was used to analyze the rice samples from different origins to distinguish the differences of metabolites in different original rice. This work is expected to provide theoretical support for the traceability of rice geographical origin, propose a new attempt to understand the change mechanism of metabolic pathways in rice samples and promote the development of the grain industry.

## 2. Materials and Methods

### 2.1. Materials and Reagents

In the experiment, rice samples of Daohuaxiang (DH), Huaidao No. 5 (HD), Songjing (SJ), and Changlixiang (CL) were provided by the National Grain Reserve (Wuxi, China) and Heilongjiang Wuchang Jinhe Rice Industry Co., Ltd (Wuchang City, China). After husking and polishing the brown rice samples, the white rice samples in each group were ground into powder. Five rice samples of the same variety were treated with the same processing methods, and the prepared white rice samples were stored at −80 °C in self-sealing bags for testing. The purity of methanol, formic acid, and ammonium acetate was of chromatographic grade.

### 2.2. Sample Processing

Milled white rice (100 mg) was placed into a centrifuge tube and added with 500 µL of methanol aqueous solution (80%). The samples were centrifuged at 1500 r/min at 4 °C for 20 min after vortex vibration. The supernatant was diluted with water until the methanol content was 53% and then centrifuged at 1500 r/min at 4 °C for 20 min. The supernatant was collected for LC–MS analysis. Methanol aqueous solution (53%) was used as the blank sample, and the pretreatment process was the same as the experimental samples [12].

### 2.3. Liquid Chromatography–Mass Spectrometry

The specific experimental conditions of LC–MS in this experiment are shown in Appendix A [7]. Orbitrap type LCMS (Q Exactive™ HF-X, Thermo Fisher, MA, USA) and Vanquish UHPLC with an Hypesil Gold C18 column were used. A novel data-dependent acquisition (DDA) approach was used for mass data collection. The top 10 precursor ions were selected from the mass spectrometry, and the product ions for each top 10 parent ions were collected by tandem mass spectrometry. The original LC–MS data of all rice samples were imported into Compound Discoverer (CD 3.1) software. Mass to charge and retention time were used to annotate metabolites from analytical results, and qualitative and relative quantitative results were obtained. The data processing part was based on the Linux operating system (CentOS 6.6) and software R (2011a) and Python (3.9).

### 2.4. Data Analysis

MetaX software was used to transform the data (including peak picking, annotation, and data normalization) [13], and PCA and PLS-DA analyses were performed to present differential metabolites. The metabolites identified in rice samples were annotated by the KEGG database, HMDB database, and LIPID MAPS database. The KEGG database was used to study the function and metabolic pathways of metabolites.

## 3. Results and Discussion

### 3.1. Principal Component Analysis

In this study, four rice samples (DH, HD, SJ, and CL) with five parallels were collected for analysis, and the metabolites of 20 rice samples were analyzed using PCA to classify the different varieties of rice. The PCA scatter plot (Figure 1) showed that the total score of the first three principal components was 62.61%, which represented most of the effective information of the original data. Thus, the PCA can be employed for rice origin analysis. As shown, the distribution areas of different groups of rice samples indicated the clear agglomeration for the samples in the same group without overlapping. Thus, the four rice samples (DH, HD, SJ, and CL) presented significant differences and could be identified by PCA.

### 3.2. Partial Least Squares Discriminant Analysis

As a supervised discriminant analysis method used in data classification and regression of metabolic groups, PLS-DA can ignore random errors and make data analysis more concentrated and accurate [14,15]. To further classify rice from different production areas, the PLS-DA model was used in this study to analyze the metabolic information of the four rice samples (DH, HD, SJ, and CL). Six groups of results were obtained. A comparison of the four rice samples and the specific PLS-DA model analysis and verification results are shown in Figure 2 and Appendix A.

In comparison with PCA, PLS-DA showed the results of pairwise comparison that was better for the comparison of variation of different groups. In comparison with CL samples, samples in the HD group were more similar, showing a larger variation in CL samples. The possible reason may be produced by the components or other physical and chemical properties. It was also indicated (Figure 2D,F) that the SJ samples were similar to DH or HD samples, indicating the similar metabolites or components of these samples.

The scatter plot of the PLS-DA scores of each group showed that the principal components of each group of rice samples were significantly different. The sample points of the two rice samples in each group were far apart and distributed on both sides of the left and right, without any overlapping phenomenon. There were corresponding regions with great differences in each group of rice samples, indicating obvious distinction. In addition, the R2Y of the discriminant combination model for different rice varieties was 1.00, which demonstrated that the original data retained reached as high as 100%. The prediction ability of the PLS-DA model Q2Y was above 0.95, and R2Y was greater than Q2Y, indicating that the established model presented good prediction ability. In the ranking verification model, all the R2 values of the six groups of ranking verification graphs were greater than the Q2 value, and the intercepts between the Q2 regression line and the Y axis were less than 0. Thus, the model without overfitting could describe the sample information with a satisfactory discrimination effect.

### 3.3. Cluster Analysis

Hierarchical cluster analysis can visually present the differences in the content of metabolites in different types of rice in the form of a heatmap [16], thereby allowing origin traceability by comparing the difference in the metabolite content in different rice samples. In the heatmap of cluster analysis, the same row represents the same differential metabolites, and the same column represents the same rice samples. Different colors represent varying contents of differential metabolites. Red indicates high content, and blue indicates low content. Cluster analysis of differential metabolites in rice was carried out and expressed in the form of a heatmap. A significant color distribution was observed, indicating that different rice samples presented obvious high-content expression and low-content expression regions that could be employed to distinguish rice origins.

As shown in Figure 3, comprehensive cluster analysis was conducted on four rice samples (DH, HD, SJ, and CL), and a total of 328 differential metabolites were determined. The content distribution of differential metabolites in the four rice samples is presented in the form of a heatmap. The heatmap with chemical information is shown in Appendix A. In DH samples, choline chloride, prostaglandin H1, 2-methoxyestrone, 17α-hydroxypregnenolone, 16α-hydroxyestrone, and other metabolites were significantly higher than those in other rice samples. High- and low-content distribution was obvious (high-content metabolites (red) were distributed above, whereas low-content metabolites (blue) were distributed below). In HD samples, the contents of 2,4-dinitrophenol, 5-hydroxyindole, histamine, 3,5-dihydroxybenzoic acid, 3-hydroxypyridic acid, 8-bromoguanosine, hydroquinone, and other metabolites were significantly higher than those in other combinations. Obvious high- and low-content distributions were observed; high-content metabolites (red) were distributed below, and low-content metabolites (blue) were distributed above. In the SJ sample, the contents of metabolites, such as N-(4-chlorophenyl)-N′-cyclohexyl thiourea, 4,5-dicaffeoylquinic acid, guanosine, 2-methylglutaric acid, and uridine, were higher than those of other combinations. High-content metabolites (red) were distributed in the middle and below, and low-content metabolites (blue) were distributed on both sides. Among CL samples, the contents of 6,7-dihydroxycoumarin, 7-hydroxy-4-chromone, L-tyrosine, 4-hydroxybenzoic acid, and other metabolites were significantly higher than those in other rice samples. This trend was roughly described in the heatmap with different distributions of high-content (red) and low-content metabolites (blue).

Therefore, the cluster analysis of differential metabolites can accurately screen the metabolites with significant differences in rice samples. By comparing the content of differential metabolites in varying rice samples, the four rice samples were successfully distinguished. On the basis of the color distribution shown in the heatmap of different metabolites, the types and contents of the four rice metabolites varied, which could be used for geographical origin identification.

### 3.4. Determination of Differential Metabolites

The screening of differential metabolites refers to the three parameters of VIP, FC, and *p*-value. VIP value refers to the contribution of metabolites. FC means fold change. The threshold was set as VIP > 1, FC > 2 or FC < 0.5, and *p*-value < 0.05. Two rice samples were selected each time for differential metabolite screening. In a volcanic map, each point represents a metabolite. The red point represents a metabolite with significantly upregulated content. The green point represents a metabolite with significantly downregulated content. The VIP value is represented by the size of the dot. The volcanic map results of differential metabolites (Table 1) of DH, HD, SJ, and CL are shown in Figure 4.

### 3.5. Annotation of Metabolites

Based on the mzcloud, mzvault, and masslist databases, combined with molecular formula prediction of molecular ion peaks and fragment ions [17], this study performed qualitative and relative quantitative analyses of rice metabolites from DH, HD, SJ, and CL. The results showed that 494 metabolites were identified in four rice samples. The KEGG database, HMDB database, and LIPID MAPS database were all used to annotate the pathways and classification of metabolites identified in rice samples. A total of 206 metabolites were annotated in the KEGG database, 220 metabolites were annotated in the HMDB database, and 115 metabolites were annotated in the LIPID MAPS database.

KEGG contains multiple databases, in which the KEGG pathway database is a collection of metabolic pathways [18]. Biological metabolic pathways can be divided into seven categories, and each category is subdivided into secondary and tertiary categories. In this study, differential metabolites were annotated by the KEGG pathway, including 7 environmental information processing pathways, 1 genetic information processing pathway, and 198 metabolic pathways (Figure 5A).

HMDB is a database containing details of small-molecule metabolites found in the human body, as well as their biological effects, physiological concentrations, disease associations, chemical reactions, and metabolic pathways. In this study, a total of 220 differential metabolites were identified in four rice samples by HMDB database annotation (Appendix A), and their information was quickly classified. Among them, the number of lipid and lipid molecular metabolites was the largest.

As the largest public lipid database, the LIPID MAPS database can annotate eight major lipid categories and their subclasses, and each category has its own next classification. In this study, 115 metabolites in rice were identified based on the LIPID MAPS database, which could be roughly divided into five categories (Figure 5B), namely, fatty acids (45), glycerol phospholipids (48), polyketides (11), proenol ketone lipid (1), and sterol lipids (10).

### 3.6. Metabolite Pathway Analysis

The pathway enrichment analysis of metabolites in rice (DH, HD, SJ, and CL) was carried out using the KEGG database. A total of 494 metabolites were detected based on the KEGG, HMDB, and LIPID MAPS databases, of which 206 metabolites were annotated by the KEGG database. In this study, two rice samples in four rice samples were selected for analysis each time. The number of background (all) metabolites in KEGG annotation was 102. The obtained enrichment data were represented by the KEGG enrichment bubble diagram, and the abscissa was x/y (the number of differential metabolites in the corresponding metabolic pathway/the number of total metabolites identified in the pathway). The larger the value is, the higher the enrichment degree of differential metabolites in the pathway is.

As shown in Table 2, six groups of analysis results were obtained (showing the results of the Top 20). Pathway enrichment analysis of rice samples was carried out based on the KEGG database, and the three main metabolic pathways with the highest enrichment degree in each group of rice samples were determined. The main biochemical metabolic pathways and signal transduction pathways involved in the different metabolites in rice were obtained. As shown in KEGG annotation results, amino acid metabolism, carbohydrate metabolism, and lipid metabolism contributed to the metabolic differences between rice cultivars. It was also concluded that lipid and lipid-like molecules are the largest different chemicals. The chemical differences were possibly produced by different genetics and different regional environments [19,20,21], and the samples in this study were collected from Northeast China and Jiangsu Province. Fatty acid degradation pathways composed of glutaric acid and palmitic acid may be a possible biological process for the different chemicals relating to the rice regions [22,23].

As mentioned above, the chemicals or metabolites of rice samples collected from different seasons or environmental conditions may be obviously different. However, this study was limited to representing the overall differences between rice samples. The different chemicals in rice samples were not equal to the different aroma or nutrition properties. Moreover, the rice samples should be collected with more biological replicates for the improvement of the accuracy of future research.

## 4. Conclusions

This study analyzed four kinds of rice samples, DH, HD, SJ, and CL, by non-targeted metabolomics technology based on LC-MS. The metabolic differences between the four types of rice samples were significant, which could be identified by PCA and PLS-DA models. The KEGG database annotated 206 metabolites, the HMDB database annotated 220 metabolites, and the LIPID MAPS database annotated 115 metabolites. The analysis of different metabolites indicated that the types and contents of metabolites in the four kinds of rice were different, presenting an alternative to determining rice’s geographical origin. In conclusion, this study provides theoretical support for developing a rice geographical origin method that can contribute to rice quality control and introduces a new attempt to understand the mechanism of metabolic pathway changes in rice produced from different origins.

## Figures and Tables

**Figure 1 foods-11-03318-f001:**
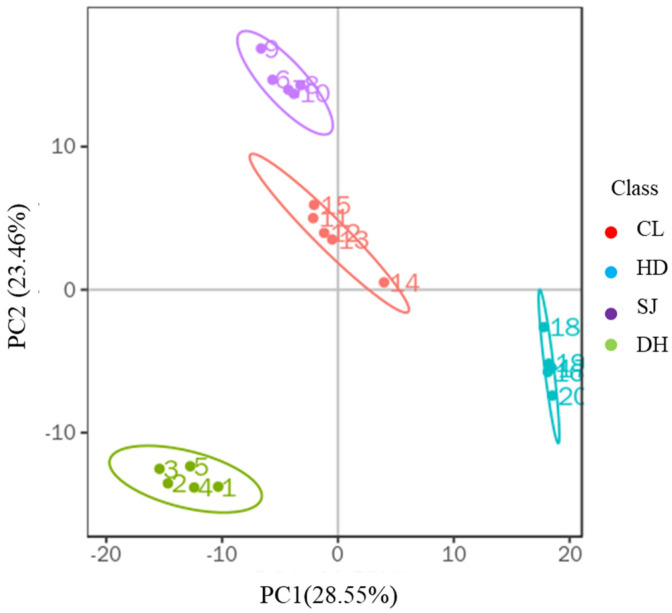
Three-dimensional diagram of PCA of four rice samples (Daohuaxiang (DH), Huaidao No. 5 (HD), Songjing (SJ), and Changlixiang (CL)).

**Figure 2 foods-11-03318-f002:**
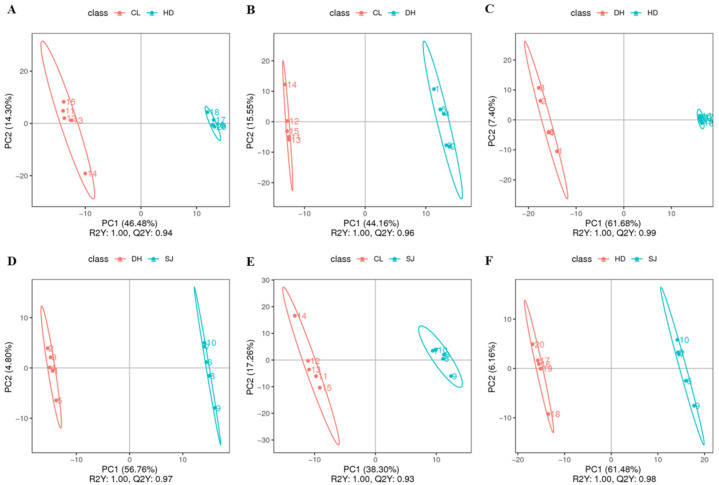
PLS-DA score scatter plots. (**A**) CL vs. HD; (**B**) CL vs. DH; (**C**) DH vs. HD; (**D**) DH vs. SJ; (**E**) SJ vs. CL; (**F**) SJ vs. HD. Notes: Daohuaxiang (DH), Huaidao No. 5 (HD), Songjing (SJ), and Changlixiang (CL).

**Figure 3 foods-11-03318-f003:**
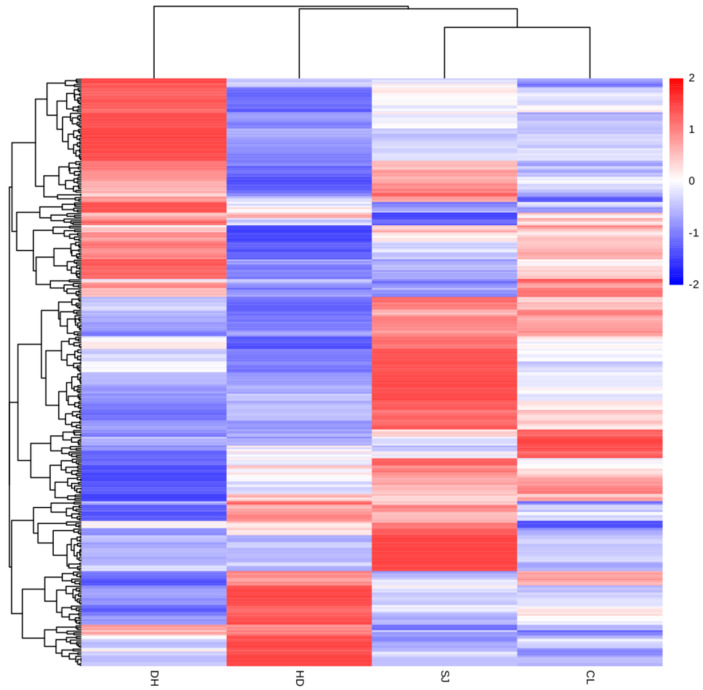
Heatmap of cluster analysis of differential metabolites in four rice samples (Daohuaxiang (DH), Huaidao No. 5 (HD), Songjing (SJ), and Changlixiang (CL)).

**Figure 4 foods-11-03318-f004:**
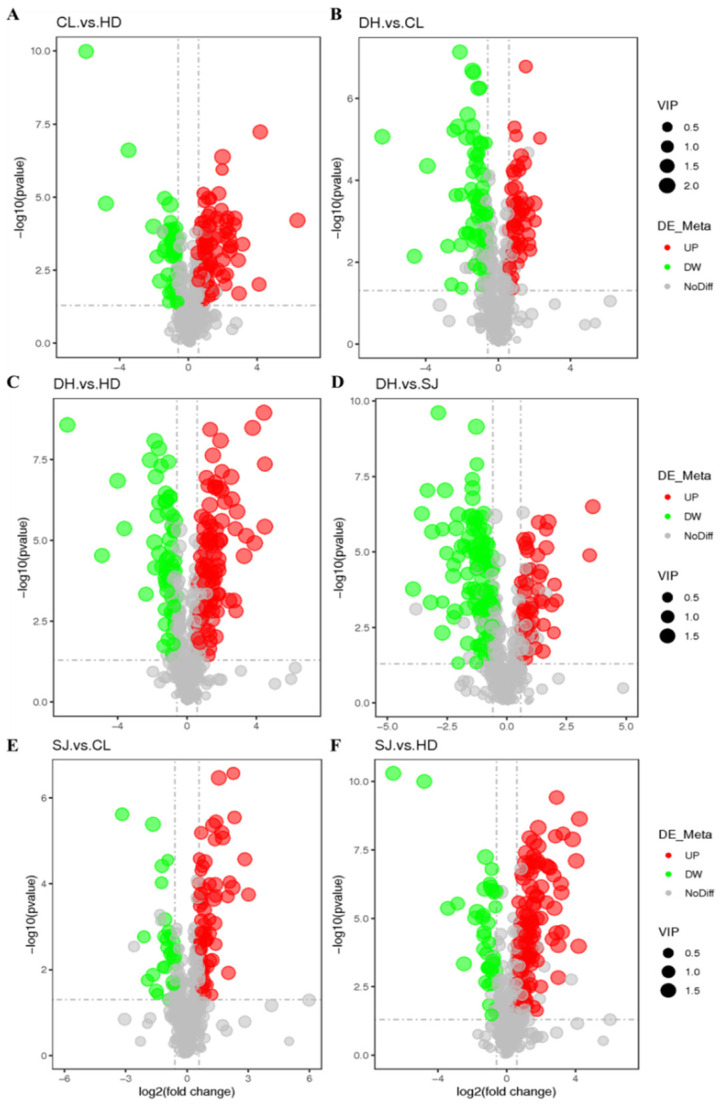
Volcanic map results of different groups of differential metabolites. (**A**) CL vs. HD; (**B**) CL vs. DH; (**C**) DH vs. HD; (**D**) DH vs. SJ; (**E**) SJ vs. CL; (**F**) SJ vs. HD. Daohuaxiang (DH), Huaidao No. 5 (HD), Songjing (SJ), and Changlixiang (CL). Note: UP is upregulated; DW is downward; NoDiff is no significant difference; the abscissa represents the change in difference multiples, and the ordinate represents the level of difference visibility.

**Figure 5 foods-11-03318-f005:**
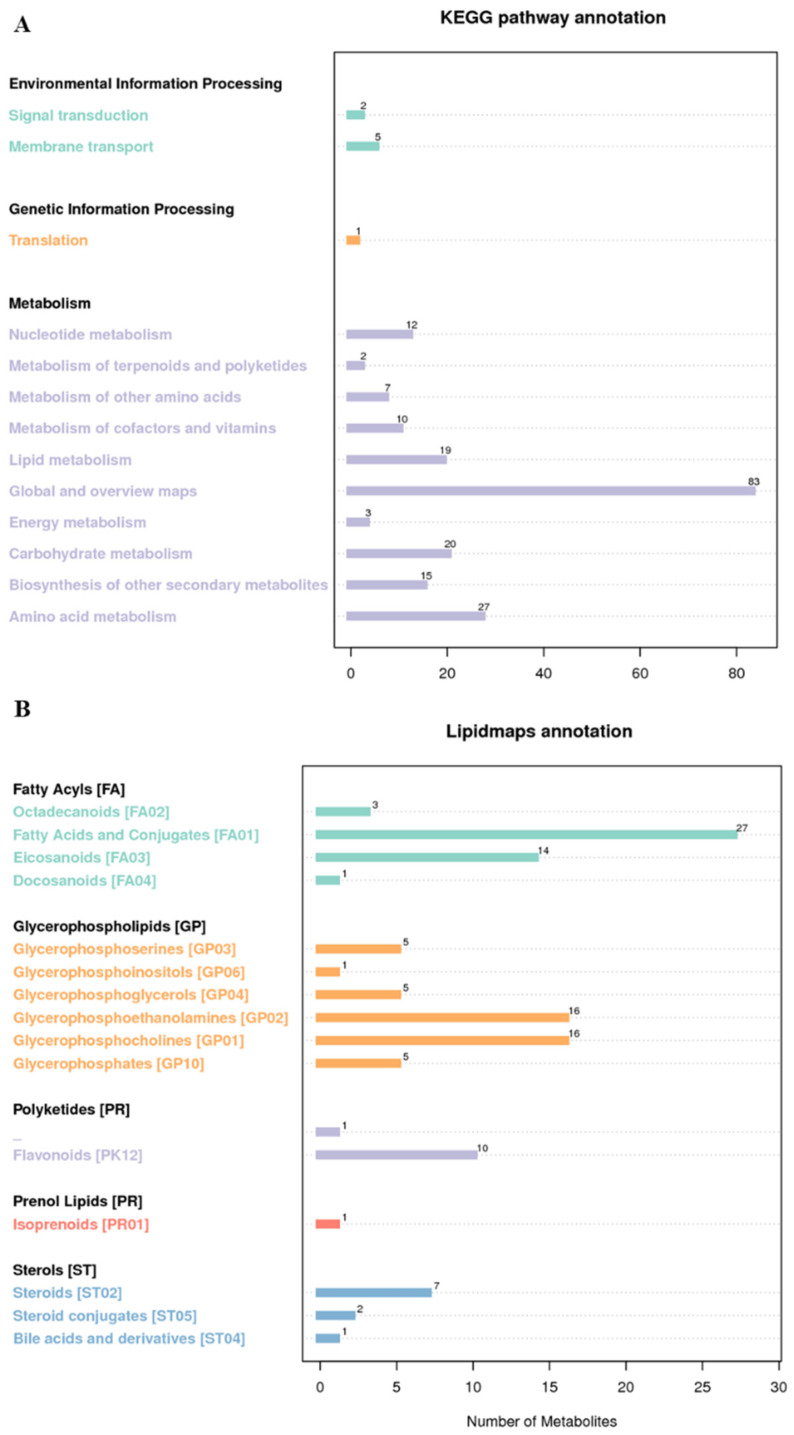
KEGG (**A**) and LIPID Maps (**B**) annotation results.

**Table 1 foods-11-03318-t001:** Quantitative identification of differential metabolites.

Group	Number of Differential Metabolites	Up	Down
CL vs. HD	133	93	40
CL vs. DH	126	54	72
DH vs. SJ	164	54	110
DH vs. HD	161	102	59
SJ vs. CL	94	66	28
SJ vs. HD	167	125	42

**Table 2 foods-11-03318-t002:** Metabolite pathway analysis of rice from different origins.

Group	TOP3 Pathways	Number of Differential Metabolites
CL vs. HD	(a) Caffeine metabolic pathway composed of xanthine and 7—methyl xanthine (b) Biosynthesis pathway of terpenoid main chain composed of valproic acid; (c) The biosynthesis pathways of stilbenes, diarylheptanes and gingerols composed of chlorogenic acid.	22
CL vs. DH	(a) Biosynthesis pathway of panquinone and other terpene quinones composed of L-tyrosine, transcinnamic acid and 4-hydroxybenzoic acid; (b) Taurine and taurine metabolic pathways composed of sulfoacetic acid and taurine; (c) Plant hormone signal transduction pathway composed of jasmonic acid and salicylic acid	32
DH vs. SJ	(a) Fatty acid degradation pathway composed of glutaric acid and palmitic acid; (b) Lysine degradation pathway composed of glutaric acid and acetic acid; (c) Taurine and taurine metabolic pathways composed of sulfoacetic acid and taurine.	39
DH vs. HD	(a) Fatty acid degradation pathway composed of glutaric acid and palmitic acid; (b) Tryptophan metabolic pathway composed of N-formyl canine uridine and serotonin; (c) Biosynthesis pathway of pantothenic acid and coenzyme A composed of 3′-dephosphate-CoA and pantothenic acid	35
SJ vs. CL	(a) Caffeine metabolic pathway composed of 7-methylxanthine and xanthine; (b) Riboflavin metabolic pathway composed of vitamin B2; (c) Vitamin B6 metabolic pathway composed of 4-pyridoxine.	18
SJ vs. HD	(a) Ascorbic acid and aldoic acid metabolic pathway composed of d-glycoacid and L-ascorbic acid; (b) Valine, leucine and isoleucine degradation pathways composed of methylmalonic acid and acetoacetic acid; (c) Propionic acid metabolic pathway composed of methylmalonic acid and acetoacetic acid.	39

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
