# Peer review of "Geographical Origin Differentiation of Rice by LC–MS-Based Non-Targeted Metabolomics"

_foods, 2022, doi:10.3390/foods11213318_

Round 1
Reviewer 1 Report
The manuscript of “Geographical origin determination of rice by LC–MS-based non-targeted metabolomics” is well prepared. This study provides theoretical support for the geographical origins of rice and elucidates the change mechanism of rice metabolic pathways in favor of improving rice quality control. There are some places which need to be revised before being accepted. Some are listed as follows.
1. Line 61 “,” after the word “samples” should be deleted. It appears that you have an unnecessary comma in a compound predicate. Consider removing it.
2. Line 77 “the” should be added before “blank sample”.
3. Line 88 “analysis” should be revised as “analyses”.
4. Line 130 “a” should be added before “satisfactory”.
5. Line 148, “the” should be added before “comprehensive cluster analysis”.
6. Line 248, “of” should be revised as “between”.
7. Line 274, the style “Oryza sativa” should be italiic. Please check the other scientific name.
8. Figure 3 was not clear. Please check it and replace it. Meanwhile, the full name of abbreviations should be given in the figure legend.
Reviewer 2 Report
Metabolomics should be used instead of the term "metabonomics". These are not interchangeable terms.
Are these genetically similar/ identical rice strains? Or are they different rice strains?
Are these rice samples all from the same year?
Were similar farming practices used to produce these rice samples?
lines 82-84 This is metabolite identification based on LC-MS analysis. "metabolic parameters" refers to biological traits, but this is not used for the chemical analysis. A more accurate way of saying this is "Mass to charge and retention time were used to annotate metabolites from analytical results".
Please indicate a little more information for analytical analysis to give reader an idea of what was done without supplemental method tables needed. What type of Mass Spectrometer was used? (TOF,Orbi,QqQ?) What type of chromatography? (RP, HILIC?). You say LC-MS, but later refer to fragmentation spectra. Please indicate LC-MS/MS if this is tandem mass spectrometry and was it all-ion fragmentation or dda?
line 88 Please indicate what transformation type was performed.
"This section may be divided by subheadings. It should provide a concise and precise 95 description of the experimental results, their interpretation, as well as the experimental 96 conclusions that can be drawn." I don't think this was meant to be kept...
line 103-106. Notes on disputable wording. How was the PCA plot "reliable"? PCA is just a method of plotting data, not a statistical test... The term "good" is not scientific. It was good in what respect? The groups were "significantly" different. Significant in what respect? statistically, biologically. these words should be scientifically justified.
Figure 2 shows much larger variation in CL compared to HD (A1), and DH compared to HD (C1). How would you explain these differences? What useful information do all these PLS-DA plots show that the PCA does not already tell us?
"The overall distribution of each metabolite on the volcanic map is intuitive and clear." I do not agree with this. I cannot get information on the "overall distribution of each metabolite" since each metabolite is only one point and thus distribution is not included in this plot.
Figure 3. Is there any way to add some sort of chemical information to this? This again tells us that the samples are different, but not much more. Are there for example groups of lipids, or other biologically related chemicals that could be indicated in the figure to add to the value of the figure?
There is a Results section and a Conclusion section, but there is not much Discussion included. Please discuss whether you think these chemical differences are from different genetics, different regional environments, or could these be from different years of production. You mention that this paper provides a new way to assess metabolic differences between rice cultivars, but did you find or can you predict or propose any plausible biologocal or metabolic phenomena from these results and the analysis? Specifically any biological explanation relating the regions would be powerful.
Please include a "limitations" section. Are these differences consistent between different years? Between different farms? The samples could be chemically distinguished, but are the samples representative of the overall class of rice and how much variation do you think there would be with more biological replicates?
Please include a link to raw data from the study for transparency and to provide the necessary information for others to reproduce or assess metabolomics claims. Freely available submission for example to the metabolomics workbench (https://www.metabolomicsworkbench.org) or any other repository to allow scientists to download data.
The title includes the word "determination", but the geographical location was not determined based on chemical data. This word should be "differentiation" based on the results/ approach and information presented in the paper.
